# At the Intersection of Transnationalism, Identity, and Conocimiento: An Autoethnography

**DOI:** 10.3390/bs15111539

**Published:** 2025-11-12

**Authors:** Isaac Frausto-Hernandez

**Affiliations:** Department of Teacher Education, The University of Texas at El Paso, El Paso, TX 79968, USA; isaacfhdz@gmail.com

**Keywords:** *conocimiento*, identity, transnational capital, transnational youth, transnationalism, transnationals

## Abstract

The transnationalism phenomenon is at the forefront of today’s globalized world. This ethnographic self-reconstruction recounts a personal story as a transnational. I reflect on my engagement in continuous transnational migration practices, and on how these have and continue to mediate my identity [re]construction and my development of *conocimiento*. These insights help not only contribute to the under-explored body of literature that has adopted a comparative stance in looking at transnationalism and education on both sides of the U.S.–Mexico border, but also seek to promote conversation around the behaviors and cognitive benefits of those who engage in such ongoing migration practices and are immersed in educational settings.

## 1. Introduction

For more than a generation, the movement of people, capital, products, information, and knowledge within and beyond nation-states has been incessant and ongoing ([128]). In current times, many people in some way or another cross the boundaries of a political or administrative unit for various purposes and varying periods of time ([11]). Rising political conflicts throughout the world are leading to people being displaced. As a result, many people decide to migrate from one setting to another, mainly in pursuit of better economic endeavors. This population of migrants includes a large number of children and youth who are to follow their parents, along with those who may be unaccompanied.

It becomes crucial, then, to further explore the transnationalism phenomena along with those who engage or have engaged in ongoing migration practices. Indeed, [2] ([2]) urged that “*necesitamos hacer teorías* that will rewrite history using race, class, gender, and ethnicity as categories of analysis, theories that cross borders, that blur boundaries-new kinds of theories with new theorizing methods” (p. xxv). Hence, exploring the deterritorialized nation-state, or the idea that economic, cultural, and political processes extend beyond the boundaries of the nation-state ([4]), contributes to an important and not-so-nascent body of literature that explores transnationalism and its relationship with and/or impact in fields such as education (e.g., [8]; [17]; [19]; [26]; [47], [48]; [61]; [64], [65]; [67]; [80]; [84]; [87]; [89], [90]; [95]; [96]; [108], [109]; [113]; [114]; [116]; [125]; [127]; [131]; [136], [137], [140], [141]). This, in turn, may better inform the current transnational realities, as well as hint at areas of interest which are worth exploring more in-depth with regard to preparing educators for our transnational present and future.

I situate my study from a personal narrative migrating among and between both Mexico and the United States. Despite the complex socio-political relationship that both Mexico and the United States share, migration to and from both countries is a common practice due to their proximity. Consequently, Mexican immigrant families and communities, living within and across two nation-states, experience both fluidity and divisions. Their lives both transcend and are separated by the geopolitical border ([6]). Thus, ongoing migration practices become habitual in the sense that they are persistent, ongoing, and continuous, although never homogeneous. They are rather unpredictable. As such, “different subjectivities, experiences, and circumstances intersect with political-economic realities, shaping who migrates, if, when, and how often they do so, and the character of their border crossings and lengths of stay in the United States” ([6]).

Early on, theorizing about transnationalism, [69] ([69]) noted that the transnational phenomenon implied a reordering of binaries; that is, the cultural, social, and epistemological distinctions of the modern period, where there was thought to be a well-established and well-marked differentiation between “us” and “them”, are less delineated as a result of transnational migration. Research around transnationalism, as well as its many factors and effects, has gained increasing attention across disciplines since it was taken up in the 1990s. Scholars in a variety of fields study the transnationalism phenomenon, including: economics ([52]), history ([53]), sociology ([45]; [99]; [120]), education ([55]; [108], [109]; [114]), and anthropology ([4]; [94]), including the sub-field of educational anthropology ([138]).

Research on transnationalism gained noticeable attention in the field of anthropology and has developed from a framework ([39]) into a research agenda ([77]) that explores the variety of migration practices from various perspectives. One body of scholarly literature has focused on how people navigate transnational lives and practices ([41]; [43]; [70]; [74]; [92]). Another group of studies has centered on particular ethnic groups engaging in transnational migration ([7]; [31]; [42]; [81], [82]; [110]; [120]; [132]; [135]), and who may seek to acculturate into the receiving culture ([40]; [44]; [97]; [100]; [122]).

Still other research explores the early transnational practices of border crossers, paving the way to deterritorialized nation-states. Deterritorialized nation-states are liminal spaces that superimpose geopolitical borders ([4]; [5]; [33]; [73]; [78]; [91]; [98]; [119]; [130]). As such, these spaces acknowledge a deep transformation of the link between one’s everyday cultural experiences and the configuration one has as a local being ([126]). Lastly, attention has been given to the effects of the transnationalism phenomenon in educational contexts ([46]; [75]; [108], [109], [111]; [123]; [133]).

Scholars have explored transnationals largely in education by looking at special topics that include the challenges transnationals face ([24]; [136], [140]), to the way they [re]construct their identities ([3]; [85]; [87]), and on how they rely on and incorporate their linguistic capital as they develop along English language teaching practices in settings such as Mexico ([86]; [96]; [104]).

The vast majority of this scholarly work has explored transnationalism at a particular moment in the lives of transnationals. Those moments include the following: arriving at an unfamiliar location, pursuing higher education, and engaging in initial teaching practices. This scholarly work has also explored the transnational in one location, predominantly the receiving country. At the center of this research is the issue of a small body of literature which has taken a comparative stance in looking at transnationalism in education on both sides of the U.S.–Mexico border ([7], [8]; [47], [48]; [108], [109]; [121]; [138], [139]). In this paper, I draw on my own story, having engaged in transnational migration practices among and between both Mexico and the United States, to retell how I pursued various degrees of education along with the various elements that I identified as crucial as I navigated different social and familial settings on both sides of the border. In doing so, I adopt a comparative stance in exploring how transnational migration practices mediate the identities and *conocimiento* (i.e., knowledge, understanding, skillsets) of those with transnational migration backgrounds. This paper first discusses an array of literature to situate transnationalism and transnational children and youth within education. I follow by describing autoethnography as my methodology. I then allude to [re]engaging in transitions, [re]creating networks, and [re]developing attachments through retelling my personal story. In doing so, I present a framework to aid in understanding transnationalism and its interrelated elements. I end the paper with a personal conclusion.

## 2. Literature Review

### 2.1. Situating Transnationalism and Education

Transnational migration experiences have been rigorously documented historically ([53]) and have recently gained further interest ([118]). Early understandings of transnationalism defined it as processes in which migrants develop social fields that expand beyond national borders, and link the countries of origin and settlement ([4]). Relevant scholarly work pertaining to transnationalism has sought to portray how many migrants have the skills, yearning, necessity, or intentionality to preserve various connections to their homes or countries of origin through social, economic, political, religious, and/or familial ties while developing and negotiating membership to the host country ([39]; [75]; [77], [78]; [91]; [98]; [119], [120]). Moreover, depending on the level of engagement with transnationalism and the strength of the ties that have been developed, households experience the movement and exchanges of goods, money, and people, as well as information, advice, care, love, and systems of power ([107]) to varying degrees, based mainly on the frequency of migration practices of transnationals from one setting to another. The flow of these elements is bi-directional, from the country of origin to the new place of settlement, and vice versa. At the core of these flows are the transnational networks and the ties that sustain these connections across borders.

Within the field of education, research specifically related to transnationals in education began to gain further momentum in the early 2000s ([55]; [108], [109]; [114]). Much of this work seeks to understand transnational students and families who reside in the United States, and paints a rather static portrait of a moment in the lives of these transnationals and their families, such as arriving in the new country or enrolling in school. Further scholarly work ([7], [8]; [28]; [76]) also seeks to understand transnational students’ and their families’ experiences, particularly with regard to how they negotiate language, identities, and broader political and economic conditions.

Some educational scholars have also focused their work outside of the United States to further inform the field of transnational educational research. More so, scholars have focused their attention on teachers throughout Mexico ([84]; [95]; [138]). Their work highlights the struggles of teachers in coping with transnational K-12 students ([138]), as well as the cultural and linguistic assets that youth with transnational experiences have as they pursue and develop in English language teaching practices throughout Mexico ([19]; [86]; [95]). This shift in focus outside the United States helps educators understand the longer effects and consequences of their work (e.g., transnational students becoming English teachers in Mexico), as well as gaining further insights into the sending country’s complex contexts and histories.

### 2.2. Transnational Children and Youth

While transnational children and youth develop linguistic and cultural understandings as a result of their migration experiences, they also face challenges on both sides of the border. In this section, I explore the literature on the schooling experiences of transnational children and youth both in the United States and in Mexico. I begin by looking at scholarly work from the United States on the matter.

Transnational children and youth throughout the U.S. educational system tend to be viewed through a subtractive ([129]), a minoritized ([36]), and/or a deficit lens. Scholarly investigation has been conducted to understand immigrant students in the U.S. K-12 setting who have beginning-level English proficiency levels and who are developing proficiency in the English language. These students are commonly referred to as English Language Learners (ELLs) ([35]), despite the previously common coding of Limited English Proficient (LEPs) students, a term used until 2015 or so ([34]). The LEP lens views children and youth as lacking linguistic and cognitive abilities that interfere with their development to advance academically. [54] ([54]) urge us to use the term Emerging Bilinguals (EBs) instead to refer to the process of language learning, rather than a particular developmental stage. The term Emerging Bilinguals (EBs) views children in the United States as those acquiring a non-English language in addition to English, simultaneously.

However, challenges arise in determining who these children and youth are. The misunderstanding of the characteristics that these learners possess, as well as the significant inconsistencies in the data used to refer to them, leads to an array of conflicts in determining who they are, along with inaccurate accounts of the numbers of this population ([34]). An additional challenge arises as this population may be moving in and out of classifications that view them through a deficit perspective. These challenges, among other social characteristics, mean that this transnational population tends to face racial discrimination ([106]).

Significant scholarly work has been conducted to aim to bridge the understanding of U.S. schooling for transnationals and transnational families. [72] ([72]), for instance, presents a long-term ethnographic study to analyze the vignettes of three transnational students. Her study seeks to understand how to better understand second-generation transnational migrants, and the findings of her work demonstrate the ways in which second-generation transnationals learn to negotiate their socialization according to the social networks they form. These networks may be initiated within the family and extended into the educational setting.

In her long-term, multi-sited ethnographic study, [66] ([66]) presents the distinct ways of knowing of four Mexican-origin, working-class families. She argues that these ways of knowing are challenged by children and new generations of these transnational families as they develop in society and within the educational setting. Moreover, schooling systems in both the United States and in Mexico fail to fully acknowledge transnational migration practices and experiences. She advocates for educators to strengthen their understanding of students and families who engage in such practices.

In their longitudinal, multiple-case study project, [18] ([18]) explore how transnational children engage in literacy practices along with their families. The kindergarten, first-grade, and second-grade students of their study developed transnational funds of knowledge from engaging in literacy practices. They also argue that these funds of knowledge should be further recognized in classrooms and schools, as they can nurture greater schooling transitions for present and future transnational children.

Additional scholarly work also aims to acknowledge transnational migration experiences shaped by the particular bilingual and bicultural identities in promoting student adaptation to U.S. schools. The work of [113] ([113]), for instance, advocates for greater work needed to help provide insight into how the lives of transnational students in the United States have been overlooked, and argues for promoting greater bilingual practices in coping with these students. The work of [103] ([103]) adopts an array of cases of transnational students in college in the United States. Their work seeks to foster student success through acknowledging the bilingual understandings of transnational students as assets.

The multi-sited ethnographic work of [64] ([64]) centers on Mexican-origin families of young females in the U.S. K-12 educational setting. Her work analyzes the challenges that develop into an underdog mentality derived from a deficit perspective towards transnational children in U.S. schooling. She suggests embracing bilingual identities as a way to counter such deficit views and, in turn, navigate within the educational setting. The work of [114] ([114]) explores the lives of Latinx transnational students in the United States who are finding their way into the educational system. Their work frames the current transnational reality and the many challenges that transnational families and students within the U.S. educational system face, such as low levels of academic achievement, being seen by teachers through a deficit lens, and feeling invisible.

The qualitative study of [21] ([21]) was carried out in the physical borderlands with young female children in elementary school. Her work sheds light on how these transfronterizx, or habitual border commuter students, can develop literacy practices through interaction with their mothers. Her work has extended to look at literacy practices of elementary students inhabiting the borderlands ([22]; [32]), as well as youth engaging in continuous border crossing experiences ([22]). The ethnographic study of [108] ([108]) centers on transnational female youth and their experiences developing in settings on both sides of the U.S.–Mexico border. She further highlights how transnational youth intentionally use different languages and literacy practices to represent themselves across borders. The implications of her work seek to highlight the sustained transnational contact with both settings, as well as to appreciate the resources that may be shared in the educational setting to promote greater academic transitions.

Furthermore, scholarly work has emphasized the factors contributing to the academic success of these students in higher education. [20] ([20]), for instance, highlight how factors such as academic self-confidence, a strong ethnic identity, internal motivation and commitment, and interactions with supportive peers hold a key role in promoting greater academic success for transnational children and youth. [56] ([56]) further provide a view into the assets of the growing number of Latinx students enrolling in higher education. Their case study work emphasizes advancing an anti-deficit achievement framework in promoting greater student outcomes. [30] ([30]) extends the conversation by highlighting how students with ongoing migration practices develop strengths and a unique consciousness as they reframe their academic experiences. Specifically, her narrative adopts an asset-based approach to looking at how Latina, first-generation youth with ongoing migration draw on their lived experiences to develop various forms of capital as they navigate through the unknown educational pipeline.

As noted, not all scholarly work pertaining to the education of transnational students in the United States adopts a deficit perspective. In fact, an assets-based approach is gradually being taken more into account in acknowledging the cultural and linguistic resources that these students possess, predominantly in Mexico. Next, I will reconsider the Mexican educational setting and scholarship about the challenges in integrating transnational students into the classroom.

The educational experiences of transnational children and youth in Mexico tend to be stratified in terms of socioeconomic status, language proficiencies, race, region, immigrant generation, school programs, and legal status ([58]). As such, scholars have shed light on the challenges that these students face as they enroll in and develop in Mexican educational settings.

An initial challenge is faced when confronting the bureaucratic process that “foreign” children and youth ought to go through while enrolling in the Mexican educational system. The Ministry of Public Education (Secretaría de Educación Pública [SEP]) recently amended regulations that required apostilles, the seals and signatures of government officials, as a way to expedite “legalized” foreign documents. In 2015, educational guidelines were amended so that school authorities would no longer require apostilles on personal and academic documents for the purposes of enrollment, providing quick schooling access for transnational and U.S. citizen students ([59]). Despite these policy amendments, many communities remain incognizant of these changes on the day-to-day level.

Analogously, [59] ([59]) emphasizes the bureaucratic challenges of Mexican-American children in attempting to enroll in the Mexican educational system in both the public and private sectors. Despite the legal amendments in seeking to eliminate bureaucratic processes, [59] ([59]) emphasizes the misinformation of such improvements, which continue to create barriers to student enrollment into the Mexican educational system. As such, immigrants seeking to enroll in the Mexican educational system continue to face a number of bureaucratic challenges due to misinformation related to legal initiatives.

A secondary challenge that the transnational population faces is that they are often invisible or non-identifiable by teachers and school officials; that is, neither their transnational experiences nor their educational needs are acknowledged. [50] ([50]) present the life stories of three children in Mexico who had previously attended public school in the United States. Their results depict the invisibility of these students as they fail to fit into the typical premises around which schooling in Mexico is organized. Likewise, [136] ([136], [139], [140]) capture the many challenges that return migrants confront in seeking to succeed within a new educational system once they have returned to Mexico, a country they do not remember. In their 2006 survey study, Zúñiga and Hamann reflect on the hidden transnational population and their poor academic results in northeastern Mexico. They extend this work as they explore the complexities of students identifying with a hyphenated identity (i.e., Mexican-American), and their identity struggles from having two national affinities ([48]). Their 2013 study describes how American-Mexican children lack agency to participate in mainstream Mexican education due to a poor understanding of the new setting, resulting in underacknowledged needs. In other words, children are unable to participate in educational and civic activities (i.e., pledge of allegiance and patriotic school celebrations) as they are unfamiliar with them ([139]). In their 2015 study, through interviews and surveys, Zúñiga and Hamann present the stories of four children and the many interrelated aspects involved in moving and adapting to the Mexican educational system. The results depict the many challenges of inhabiting a territorial dislocation by children who migrate to Mexico, the country of their birth, for the first time ([140]). In other words, these children do not return; rather, they are relocated to a new place due to the circumstances of their vulnerability, a space that is highly unfamiliar to them. These children, then, are confronted with a semi-familiar culture and its language, along with unsupportive educational policies that do little to acknowledge them.

[117] ([117]), for instance, center their case study on youth and children who enter the Mexican educational system along the borderlands. Through interviews, their work signals that the lack of support from educational practices results in the exclusion of these students from the mainstream classroom. Moreover, the transnational student population in Mexico is gradually increasing. They are likelier to attend rural schools than the mono-national Mexican student population, and on average, perform below those in urban schools, as well as those in private schools, on measures of academic performance ([102]). However, scholarly discussion has also covered the processes of transnational students’ adaptation to urban schooling as well. [105] ([105]) portrays what it is like for two transnationals to go from an urban U.S. setting to a metropolitan context in central Mexico, along with the conditions for social and labor re-insertion. Her comparative results of reinsertion with recurring migration and reinsertion with more permanent settlement suggest that a more permanent settlement is likely to result in greater investment into adaptation, while more continuous migration practices entail less investment, and hence, greater challenges.

The work of [9] ([9]) extends the conversation and explores how American-Mexican youth face acculturation challenges in rural, central Mexico. Their ethnographic study centers on three youths, who, despite their linguistic and social insecurities, are able to develop their civic identities and sense of belonging through engaging in academic endeavors. The three youths are ultimately able to rely on sociocultural resources such as engaging in communication with the surrounding society and developing agency in aspiring to belong to their receiving community; that is, belonging is revealed to be a sociocultural practice that is negotiated through relations with one another (i.e., engaging in weekly honoring of the Mexican flag) ([9]).

These challenges further increase as transnational children and youth lack the proper linguistic skills to understand class content taught in the Spanish language ([93]), as well as intercultural communication skills and norms ([49]) and the literacy skills to read and/or write in a class taught fully in Spanish ([25]). There is little academic support from the Mexican teachers as the Mexican education system, unlike the United States, lacks a Spanish as a second language program curriculum for students who have under-developed Spanish proficiency. Most educators have relatively low competence with the English language, and do not use it instructionally ([83]). Also, educators are largely unfamiliar with the U.S. education system and curricula, which contributes further to schooling complexities and higher instances of failure. The work of [137] ([137]) furthers this conversation and illustrates how the lack of preparation by Mexican educators in the Mexican educational system leads to little to no support for transnational children and youth entering their classrooms. These challenges, amongst others, inhibit them from having successful schooling experiences in Mexico. Nonetheless, a number of Mexican-supported efforts have sought to gradually build Mexican teachers’ readiness for transnational students ([51]; [115]; [142]).

Furthermore, the work of [24] ([24]), along with other colleagues ([25]; [27]), explores the challenges of transnational children in overcoming monolithic perspectives of educators in Mexican schools, or perspectives of uniformity in the Mexican educational setting. Despagne’s work places emphasis on how language and semiotic resources help transnationals adapt to the Mexican educational system. Her 2018 qualitative study, for instance, looks at how 20 transnational children integrate into Mexican schools in central Mexico, while developing their process of becoming Mexicans in an elementary school. The results suggest that unfamiliarity with institutional expectations results in silencing and social exclusion.

In a similar study, [60] ([60]) draws on surveys, focus groups, and individual semi-structured interviews to emphasize how return migrants recall their experiences in the United States to pursue higher education in Mexico. Despite being “othered” due to their English proficiency, transnational youth recall their linguistic knowledge to initially make a vocational choice in pursuing language teaching, to then create communities within their professional setting ([60]). Results show how gradual Spanish proficiency helped along the processes of socialization, inclusion, and school learning.

The narrative study of [26] ([26]) depicts the multiple ruptures (e.g., social, familial, cultural, economic) of 19 returnee youths in the process of integrating into the Mexican public high school system. Through their linguistic abilities derived from their bilingualism in the English and Spanish languages, however, the youths were able to develop their agency in repositioning themselves into the receiving community. In this particular study, the gradual proficiency of the Spanish language allowed them to become active members of their school community. Similarly, the narrative study of [27] ([27]) explores how a group of high school youth negotiate their sense of belonging to the United States and Mexico simultaneously due to their participation in educational settings in Mexico and maintaining their English proficiency in reclaiming their U.S. citizenship. Results show how a broader communicative repertoire allows youth to reclaim their citizenship in both countries. [57] ([57]) extend the conversation as they explore how young adult learners experience and learn to build their citizenship in Mexico. Their study implements focus groups to explore how 13 young adult learners who pursue higher education in Mexico develop acts of resistance and a voice that allows them to bridge tensions associated with their dual citizenship ideologies. Results show gradual engagement with broader society and with their educational environments as counteracting prejudice and deficit mindsets towards non-traditional Mexican students in Mexican educational settings.

Along with studies conducted with return migrant children and youth in Mexico, there is also scholarship from an assets-based perspective that looks at this rapidly growing transnational population and refers to them as the “students we share” ([37]; [38]). This work centers on the educational processes of U.S.-born children of Mexican migrants who now live and attend school in Mexico ([58]; [62]; [61]; [79]; [136]) and focuses on educational practices such as community-based service learning that may be considered to enhance their linguistic knowledge and needs ([124], [125]).

Not all academic discussions concerning the education of transnational children and youth in Mexico center on the challenges that this population faces. Gradually, an approach based on the perspectives and richness of their experiences is given consideration ([103]). This work helps us understand the various complexities that transnational children and youth face while engaging in transnational migration and enrolling in the educational systems of both the United States and Mexico.

### 2.3. Transnationals and Identity

As transnationals engage in continuous migration practices, they shift systems of classification, which may result in the negotiation and reconstruction of their own identities. Scholarly work has been done on how transnationals develop, construct, and reconstruct their identities. In this section, I discuss scholarly work on the connection between transnationalism and identity [re]construction.

[28]’s ([28]) multi-year ethnography highlights the fluidity of identity of a Guatemalan woman who grew up between both Guatemala and the United States. Ek’s work centered on the multiple ethnic and gender socializations experienced by the participant. Her identity ultimately shifted toward the American culture as she pursued her educational and professional endeavors in the United States, while simultaneously maintaining cultural practices that required the use of Spanish, such as being part of a Hispanic church ([28]).

[76] ([76]) seeks to offer a heterogeneity of identities around those usually referred to as “Latino”. Her long-term ethnographic study illustrates how multilingual indigenous families from several Latin American countries struggle and maintain indigenous languages and cultural practices despite discriminatory practices against indigeneity. To do so, these people maintain active engagement with their home countries through supporting economically and engaging in their hometown elections ([76]).

The conversation about transnational identities has extended to encompass basic education throughout Mexico. [3] ([3]) explore the experiences of six children who are return migrants in Mexico and the aspects involved in their identity construction through an ethnographic approach. Their study analyzes the sociocultural assets and sociolinguistic practices that transnational children use in reconstructing their identities, in the context of the Mexican educational system. One of the ways they did this was through engaging in extracurricular activities related to their English courses.

[85] ([85]) adopted a sociocultural perspective to focus specifically on the ethnic/cultural facets of the identities of transnational young adults in higher education in Mexico. Their narrative study explored the ways in which 11 transnationals reconstructed their hybrid identities in light of discriminatory challenges they faced within their ongoing migration practices, such as schism and a prejudice based on their linguistic choices. The participants embraced their backgrounds as resources to view opportunities for their future selves. They also managed to find opportunities for development upon being stigmatized due to their linguistic abilities, as well as the migratory practices they referred to. These include speaking in a particular way to fit in with specific social groups and mediating their relationships with one another to better adapt to a given location.

The scholarly conversation has also extended into the lives of transnational youths and young adults and their development in higher education in Mexico. The work of [19] ([19]), for instance, explores how transnational young adults engaged in and developed agency within the context of higher education in a Mexican university, after having been deported from Arizona, United States, to the neighboring state of Sonora, Mexico. Through implementing in-depth interviews with four transnational university students in a TESOL program, [19] ([19]) depict how they navigate the higher education pipeline in Mexico. Their work places emphasis on the legal challenges of being undocumented in the United States and the process of deportation, along with being blocked from accessing local higher education due to state laws. The study suggests that stronger ties on both sides of the border would help build agency to navigate college experiences, and ultimately determine how obtaining a university degree related to teaching English would not aid in returning (or not) to the United States.

Similarly, [87] ([87]), [89] ([89]), and [134] ([134]) explore the ways that bilingualism shapes the process of identity construction for transnational youth and young adults. The narrative work of [87] ([87]) presents the life stories of three transnationals at different stages in their educational and professional development in Mexico. Their results suggest a connection between transnational experiences and levels of investment in the process of identity construction within social, cultural, and professional involvement. Thus, the participants’ own interpretations of their lived experiences allowed them to develop varying degrees of involvement within their new societies, including the academic one. Moreover, embracing their bilingual identities led to their willingness to share their bilingual knowledge with their students, which fostered greater English usage and acculturation in the United States.

The qualitative work of [89] ([89]) presents narratives of three undergraduate participants and the ways they use language as a means to reconstruct their identities. Her results illustrate the complex interrelationship between transnational migration practices and discursive practices, resulting in a particular metalinguistic consciousness of how to adjust to different social settings. In turn, the participants developed translanguaging practices that resembled their backgrounds, and simultaneously, they were able to adjust to different domains.

[88] ([88]) extends the conversation by exploring the role of remittances in identity construction for transnational undergraduates and pre-service teachers in North-Central Mexico. For her, remittances are not only economic, but also social and cultural practices. They also involve ideologies that may be transmitted from one country to another. The results of her narrative study show how sociocultural remittances, such as traditional values and physical goods, help transnationals develop their worldviews as well as promote ties and engagement on both sides of the U.S.–Mexico border, and ultimately impact their personal identity construction.

The work of [134] ([134]) similarly illustrates how intentional linguistic variations within the English language are crucial in constructing social identities, and ultimately learning and transmitting cultural knowledge. As such, these intentional language variations serve to indicate bilingual membership. Examples include omitting the word-final “-s” for Central Americans, adapting verb conjugations from African American Vernacular English (AAVE), such as omission of the copula be for Puerto Ricans, and mixing both English and Spanish in what is known as Spanglish (or Tex-Mex) for those of Latinx descent.

[14]’s ([14]) study reports on how transnational students employ various languaging practices (such as mixing and blending languages and/or translanguaging) in promoting a particular transnational identity on social media. Similarly, the results of her 2015b ([15]) online ethnographic study show how five transnational youth used linguistic features to construct their ethnic identity on social media, highlighting their Mexicanness as embraced cultural features. For these participants, a sense of mocking the Spanish language (e.g., using *mula* for car, or **guaay* [*güey*] for fool) helped them portray a *ranchero* identity online. Furthermore, [16]’s ([16]) study closely resonates with the afore-mentioned studies, emphasizing the role of digital communication in maintaining and portraying a Mexican-heritage identity. Though not necessarily mocking the Spanish language, certain linguistic variations found in traditional Mexican households (e.g., such as mispronouncing or improperly writing a word, as in the case of “**pos*” to refer to “*pues*”) allowed the participants to integrate into Mexican society. Lastly, her 2018 computer-mediated critical discourse analysis (CDA) study looks at how language renders a degree of symbolic power, and using a particular variety leads to entrée into various social groups. Her results demonstrate the power of language use on social media for developing membership in transnational communities. Examples of this are a transfer and adaptation from one language to another, as in the case of the word “*imboxeo*” to refer to an inbox message (constructed through mixing “inbox” with “*correo*”), and phrases such as “*compraré los* tickets *el* Wednesday”, which seek to construct membership to transnational communities on social media through translanguaging within a single sentence.

Additionally, scholarly work has explored initiatives to ameliorate identity complexities among transnationals, as in the case of [68]’s ([68]) work, which contests negative views of the Mexican-origin transnationals as traitors (i.e., *Malinchistas*) to their own culture. Their critical, ethnographic study depicts the stigma and prejudice that transnational youth face when migrating to their parents’ country. Their results argue for a new perspective, which invites one to appreciate the assets that transnational youth promote in an increasingly multicultural world, primarily their bi/multilingual richness, along with their ease in negotiating their identities with various social groups and domains.

Transnationalism is tied to the [re]construction of identity, and language use becomes a key element in the process of identity [re]construction. Language allows both inclusionary and exclusionary (or discriminatory) practices, and informs how transnationals decide to present themselves in academic and social settings. The academic literature also suggests a connection between lived experiences and levels of investment in the process of identity [re]construction and involvement in social, cultural, and professional involvement. As such, the goods, information, practices, and understandings that are shared from one setting to another hold a strong impact on the development of the identities of transnationals.

### 2.4. Transnational Knowledge

Due to their lived experiences engaging in ongoing migration practices, transnationals develop their knowledge, understandings, and worldviews as they maneuver throughout the world. This section analyzes scholarly work looking into the broader knowledge that transnationals develop as a result of their lived experiences.

[109] ([109]) examines the lives of three transnational Latinas from California who maintain close ties to Mexico through continuous family trips. Her ethnographic study describes out-of-school learning derived from engagement in oral literacy practices and farm work activities with their elder caretakers (often grandmothers). These practices enable these three teens to shape their identities as dual citizens. Similarly, [109]’s ([109]) ethnographic work derived from the afore-mentioned study centers on how the literacy practices of transnational teens may provide linguistic and cultural resources for navigating home bases on both sides of the U.S.–Mexico border. Examples include storytelling and gaining advice from stories told by their elder relatives. The results emphasize how family and community narratives hold a key role in providing linguistic and cultural resources, maintaining contact with both settings, and developing a sense of belonging to both locations.

[112] ([112]) also explores cultural norms and practices that Latinx youth maintain due to continuous return trips to Mexico. Her extensive ethnographic study examines the lives of three second-generation transnational youths who grew up in the United States, along with their ties to their parents’ homelands in Jalisco, Mexico. The findings suggest that local and family knowledge help develop counter-stories of resistance and assimilation into the receiving community. This includes intimate relationships with family members and religious practices that help in authenticating a Mexican identity, and thus countering discrimination in the United States. In other words, the youth in her study adopt the knowledge shared by their families to develop counter stories and, in turn, challenge full assimilation as they embrace their ethnic identities.

[96]’s ([96]) study places emphasis on the particular consciousness developed by teachers inhabiting the physical U.S.–Mexico borderlands. Using their own lived experiences in terms of engaging in ongoing migration, as well as living in the physical U.S.–Mexico borderlands, these teachers develop an understanding of better ways of maneuvering through their lives on both sides of the border. This includes, but is not limited to, using more colloquial language and common American idiomatic expressions when crossing the border, in order to cross with fewer problems. It also involves knowing where it is more beneficial to do certain paperwork/registration on one side of the border (i.e., registering a vehicle), and knowing which side of the border is better in terms of purchasing certain items (i.e., buying clothing in the United States, and buying groceries in Mexico). The authors highlight the cultural and linguistic capital acquired by these transnational teachers and their goals of sharing their epistemologies with their students.

[63] ([63]) presents a critical ethnography of four Mexican-origin families in the United States. Emphasizing the *Nepantlera*, knowing (i.e., in-betweenness) that these families developed through their ongoing migration practices amongst and between both the United States and Mexico, the author explores the ambiguities of living in the in-betweenness, as well as striving to build bridges through practices that allow these families to keep in contact with the home community in Mexico. These actions become key in allowing the participants to understand the world through their liminality ([63]). In this sense, fluidity becomes a core feature in the daily lives of the participants.

[64] ([64]) presents the case of a Mexican transnational youth in the United States by highlighting the importance of family support in adapting to the new host culture. Her ethnographic work provides insights into how four families adopted a *sobrevivencia* (i.e., survival underdog mentality) mentality as they strive to adapt to the U.S. culture. The underdog mentality alludes to a rather gritty mindset in persisting and thriving despite numerous forms of oppression ([64]). Her findings suggest that reciprocity in understanding the challenges of immigrant families among the U.S. educational system is likely to result in greater educational outcomes.

Drawing on her 2014 work, [65] ([65]) shares a multi-sited, three-year ethnographic study with four working-class Mexican-origin families. The findings of her study are examples of chained knowing, which refers to the knowledge that transnational families develop from the links and connections made and maintained with their families and communities across the U.S.–Mexico border. This also encompasses being chained or linked to family members on the other side of the border, including those who migrated before. For [65] ([65]), then, the border trickles into how families understand and look at the world; chained knowing links family members to immediate and extended relatives, and ancestors, where each influences the other.

[66] ([66]) further extends her 2014 work and expands on transnational ways of knowing as a theoretical lens in looking into understandings that emerged from four Mexican-origin families in the Washington D.C. area. Her work emphasizes the liminality of living in the “in-between” of families who are also chained to their Mexican communities and families. For [66] ([66]), *conocimiento* is at the intersection of *Nepantlera* knowing, chained knowing, and *sobrevivencia* knowing. As such, transnational families develop their understandings from living in the “in-between”, maintaining ties to their homelands, and developing an “underdog” mentality in overcoming the struggles that arise in the process of adapting to a new location.

The lived experiences of transnationals link directly to how they develop their knowledge, understandings, and worldviews. For some, the family and community embrace certain knowledge and understandings that are shared and passed on from generation to generation. For others, the many challenges that are faced when engaging in ongoing migration lead to embracing their own survival, which ultimately develops into ways of knowing and understanding the world. For others, living in an “in-between” state of mind constructs liminal spaces, understandings, and perspectives of viewing the world. Whatever the case may be, lived experiences transition into rich *conocimiento*.

### 2.5. Transnationalism and Education: A Comparative Stance

The comparative stance on transnationalism and education that I take is with the purpose of comparing and contrasting the lived experiences of transnationals on both sides of the U.S.–Mexico border. Given that I use a comparative stance in exploring the identity [re]construction and development of *conocimiento* of transnationals on both sides of the U.S.–Mexico border, it is of utmost relevance to explore research on transnationalism and education that has been done comparatively.

[7] ([7], [8]) highlights how Mexican immigrant youth, located both in the United States and back in Mexico, understand and share information about schooling through social networks. Through ethnographic methods and interviews, she centers on the perceptions of negative racial confrontations in U.S. secondary schools. She notes that family members see this as impacting transmigrant learners negatively, and they urge the promotion of a more inclusive environment for all. She also highlights the difficulties of learning English, as well as their positive outlooks for services offered in schools and caring teachers ([8]).

[138] ([138]) investigate the experiences of transnational children and youth on both sides of the U.S.–Mexico border. Their comparative, mixed-methods study explores the schooling experiences and perspectives of this population both in Mexico and in the United States. The results depict greater flexibility, greater infrastructure and accommodations, and a more relaxed schooling atmosphere in the United States. Whereas in Mexico, students had smaller and less well-maintained facilities, a more challenging curriculum, and a more rigid social atmosphere ([138]). Transnational children and youth also expressed challenges with racism in the United States, and a lack of sense of belonging while attending schools on both sides of the border.

[47] ([47]) depict the estrangements that transnational children can face on both sides of the U.S.–Mexico border as a result of continuous migration practices. Their qualitative work centers on describing the ruptures (i.e., familial and social) that children may be exposed to and may need to routinely negotiate due to the bumpiness and unpredictability of their transnational migrations. Their results emphasize the unexpected separations of families, as well as the cultural shock and mixed beliefs Mexican educators may have in dealing with these newcomers in the classroom. These beliefs include mixed feelings about the national identity being promoted, and the lack of acknowledgement of the hybrid selves being portrayed through quotidian practices and mixed language usage. [139] ([139]) extend their work by exploring the challenges that American-Mexican children face, predominantly in lacking agency to participate in mainstream education due to unfamiliarity with the new setting. Unsurprisingly, they note that this population also faces similar struggles in incorporating themselves into the American educational system.

The work by [121] ([121]) presents a multi-sited qualitative research project in the United States and Mexico. Her work sheds light on the migrant experiences, aspirations, anxieties, and expectations that young migrant women have when engaging in migration practices. Her comparative study highlights the patriarchal role of parents in bringing their children to agricultural, central California, as well as the expectations and uncertainties of the young girls and their families who remain in rural Michoacán. The transnational diaspora becomes the bridge in connecting both settings together, as well as the key source of social, economic, and cultural remittances ([121]).

## 3. Methodology

### 3.1. Autoethnography

My inquiry takes the form of an autoethnography ([13]; [29]). In broad terms, an autoethnography is “a text which people undertake to describe themselves in ways that engage with representations others have made of them” ([101]), and is a research method that is qualitative in nature, which uses a researcher’s autobiographical experiences as primary data in order to analyze and interpret the sociocultural meanings of such experiences ([13]). This form of research is conducted and represented from the point of view of the self, particularly through studying one’s own experiences. Moreover, autoethnography values the self as a rich repository of experiences and perspectives, which may not be easily available to other, traditional approaches ([10]). Furthermore, this approach recognizes that knowledge is based on one’s location and identities through engaging with the situatedness of one’s experiences ([10]).

A core purpose of autoethnography is to highlight how culture shapes and is shaped by the personal. In doing so, one’s experiences and development are socially constructed. There is a developed agency in articulating one’s own experiences rather than allowing others to represent them. As such, I draw on multiple sources (i.e., personal writing, narratives, written artifacts, etc.) as I explore some of my hidden feelings, suppressed emotions, and often forgotten memories. I further draw on elements of analytical autoethnography ([1]) to engage my narrative with theories and research findings. I also appeal to elements of interpretive autoethnography ([23]) to give space to and understand my personal stories and experiences as set in social and cultural contexts.

The story portrayed in this article is based on my personal background engaging in transnational migration from an early age. My story is molded by the global and the local, the different settings and locations I have inhabited, the people I have engaged with, and the different perspectives I have come across within academia. I engage with these multiple sources to explain my experiences and interrogate their positions. This story does not center solely on me. There are implications to enhance the language, education, and identities of transnational youth.

### 3.2. [Re]Engaging in Transitions, [Re]Creating Networks, and [Re]Developing Attachments: My Story

I was around the age of four years old when my family first decided to migrate from central Mexico to the United States. We arrived at an unknown setting, where everyone and everything was highly unfamiliar. Being immersed in a predominantly English-speaking setting in the Pacific Northwest of the United States, the local language, the culture, and its people inclined us to hold back from even the smallest form of intercultural communication. The need to run a number of errands to survive gradually pushed us to begin to use the English language as the main form of communication with the many locals of the place.

I began formal schooling once in the United States, starting from kindergarten. In a program that fostered full English immersion, I was forced to assimilate to the language. I quickly began picking up a few words of the English language, which gradually transitioned into full sentences and ultimately into full conversations. I had the support of my two siblings, with whom we all shared and learned the language from one another. The situation at home was a strict “Spanish-only” policy. My mother always urged us to use Spanish at home, often reprehending us when we used English. She continuously reinforced that we would one day go back to Mexico, and she wanted us to hold a fluid conversation with our grandparents, “without the accent” that other migrants had. We found great support from the Mexican-origin population from nearby in the neighborhood. Our languaging practices allowed us to switch, mix, blend, and shuttle to and from both English and Spanish.

At the age of ten, my two siblings and I had the opportunity to visit our grandparents in our hometown in Guanajuato for a whole year. Once in Guanajuato, I enrolled in the fifth grade of elementary school, my sister in the third year of middle school (or *secundaria*), and my brother in the second year of middle school. I recall initially being singled out for being “the foreigner”. I faced some prejudice, and soon came to realize that I also faced a number of micro-aggressions, such as being called on to read in Spanish out loud due to having a “funny accent”. Nonetheless, my mother always encouraged us to do well in school, which I responded to in a positive manner. After this year, my two siblings and I returned to our home in Oregon.

I began sixth grade of middle school once back in Oregon. I had a number of friends from my childhood, with whom I could rely on to develop a greater sense of belonging. The English language was not much of an issue, and I had a general understanding of how the educational system worked in the United States. As I did in Mexico, I also pushed to excel with my grades in the U.S. educational system. I began engaging in sports, and soon encountered a number of friends from my childhood. In the middle of seventh grade, my siblings and I moved back to Mexico for approximately a year and a half.

Once in Mexico, I enrolled in the remainder of the first year of middle school (or the equivalent of the seventh grade) and the full year of the second grade (or the equivalent of the eighth grade). I had no notion of what school was like because I enrolled in the same school that my two siblings had previously attended. In our relatively small hometown in Guanajuato, I encountered many of my elementary school friends, which helped the transition to flow smoothly. The Spanish language was no longer a barrier, as my mother continued to maintain the “Spanish-only” policy at home while in the United States. I can recall the extensive hours of class in middle school and the many subjects that were offered throughout the week. I also recall how my English teachers relied on me as an aid, helping them recall vocabulary and helping my peers as needed. I also became a broker for peers who had limited Spanish proficiency. After a year and a half in Mexico, we moved back to Oregon.

Once in Oregon, I enrolled in the freshman year of high school. Again, I had a notion of the schooling system due to previous experiences with my siblings. I continued to engage in school sports, and continued to come across many of my friends from my childhood and early adolescence. My proficiency in both Spanish and English was fluent, and I developed great confidence in speaking in both languages. In terms of my grades, my mother continued to urge us to excel, to which I continued to respond accordingly. A close relative on my mother’s side of the family became ill, and she wished to go to Mexico to help care for him. I was her only child who consciously wanted to follow her on her uncertain stay in Guanajuato.

It was winter break of my sophomore year in high school in Oregon when my mother and I traveled to Guanajuato to help care for her ill relative. This time, I was alone without my siblings, and the time of stay was uncertain. I tried to enroll in the corresponding school year of *preparatoria*, but was faced with the unfamiliar registration and enrollment processes due to the requirement of apostilles. The processing time took a few months, which ended in a lost school year, and the decision to enroll in *preparatoria* from the beginning. I enrolled in the first semester of *preparatoria* the following fall. To my surprise, I once again encountered many of my former peers and friends. This transition was quite smooth, despite the absence of my two siblings. I continued to excel in school and even earned a scholarship due to my high grade point average. Despite the quick recovery of our ill relative, my mother decided it was best that I finish my *preparatoria* studies in my hometown in Guanajuato.

Towards the end of my *preparatoria* completion, I began to look for college opportunities throughout Guanajuato, as I was cognizant of the higher cost of higher education in the United States. With the support from my siblings still in Oregon, along with that of my parents in Guanajuato, I began to apply to different college educational programs throughout Guanajuato. Coincidentally, I became cognizant of a bachelor’s degree in TESOL/ELT (Teaching of English to Speakers of Other Languages/English Language Teaching), which sought to prepare English teachers in Guanajuato city. I set my other potential admission processes aside to focus solely on this opportunity as I felt the urge to “do something” with my English linguistic capital. I went through the admission process and was admitted into the program.

Early on in my undergraduate study of TESOL/ELT at Universidad de Guanajuato, I noticed that there were indeed many other undergraduate students like me. A high percentage of us had migrated between the United States and Mexico, some on a continuous basis, and our similar experiences resonated deeply. We talked about our perceptions of the surrounding societies and the particular way of understanding the world that we had developed. Strong bonds developed into close friendships, as well as academic partnerships and networks. Unconsciously, many of us shared similar ways of viewing the world and advocated for similar academic approaches (i.e., acknowledging the students’ backgrounds and highlighting the cultural and linguistic capital developed from students with similar backgrounds). Also, many of us recalled our migration experiences within our English-teaching practices and shared those experiences and understandings with our students. We would tend to talk about how life was different in both settings (the United States and Mexico), as well as using more colloquial language on a habitual basis, which many times deviated from the standard textbook being used. There was something important about our group of English pre-service teachers, our migration practices and experiences, our identities, our understandings, and how all of this was embedded into our English-teaching practices. I realized that it was important to explore these experiences in an academic manner.

I soon became engaged with the topics of transnationalism and transnationals. I began to explore these topics more in-depth, trying to learn as much as possible about the matter. I sought to understand how the transnationalism phenomenon impacted people’s lives. I wanted to highlight their experiences, their voices, and their identities. I found support as a research assistant in my undergraduate degree and entered the world of academic research and publishing. Near the end of my undergraduate degree studies, I published my first scholarly manuscript, which dealt with transnationalism.

This publication led to my first national academic conference in Mexico. My passion for these topics grew and led me to further learning opportunities at local, state, and national level conferences, seminars, and workshops. This, in turn, allowed me to pursue a master’s degree in applied linguistics in ELT, also at Universidad de Guanajuato. I continued to develop along the research path and published more on the subject.

After earning my master’s degree, I continued to teach English and transitioned into a lecturer role at Universidad de Guanajuato. Soon, I became involved with curriculum design and the admissions committee for a number of undergraduate programs at the university, which led to my working with transnational students pursuing undergraduate degrees. As I advocated for other transnational students, I gained a better understanding of the transnationalism phenomenon and those who engage in such ongoing migration practices. Understanding the many assets that transnationals bring into the educational setting drove me to pursue a doctoral degree in Teaching, Learning, and Culture at the University of Texas at El Paso. Through my dissertation study, I sought to better understand transnationalism, what it entails, and what the practices are, as well as highlight the many assets that transnationals who become teachers may bring into the classroom as a result of their experiences. Combining my personal experiences engaging in transnational migration practices with scholarly research on the lived experiences and identities of transnational teachers has been key to the conceptualization of my research.

### 3.3. At the Crossroads of Conocimiento, Identity, and Transnational Capital

In reflecting on my own life story and those of other individuals who have engaged in migratory movements among and between two different geographical settings and cultural contexts, I find it useful to also draw on interrelated ideas emerging from the literature on transitions that occur through the life course of a given person. I draw on the work of [87] ([87]) to propose a model of transnationalism and its interrelated elements, which may ultimately impact the language, education, and identity development of transnational children and youth, as well as their social and professional involvement with their surrounding society/societies.

Transnationalism entails facing a number of challenges in adapting to a new setting. Whether being prepared for a physical transition or not, transnationalism comes with doubts and uncertainties, and denotes being exposed to a new and unknown physical space, unknown people, and often an unknown language as well. Transnationalism also implies a negotiation of a sense of attachment to a location, its culture, and possibly its language, which may result in emotional scars from becoming detached from the “home” familiarity. Furthermore, transnationalism denotes strong kinship among those who engage in such practices. In turn, engaging in transnationalism signifies a [re]construction of identity and *conocimiento*.

Figure 1 is an attempt to illustrate my understanding of transnationalism. The figure encompasses a number of synergic elements that cross-pollinate each other and ultimately lead to the development of ways of knowing, which in turn impact involvement with the world. The figure is described below.

The upper section of the diagram presents the interrelated elements of transnationalism. The diagram presents a number of continuums, in which a person can be placed according to inclination to one aspect over the other, or rather equally to both. The incorporation of various continuums is done in an intentional manner to posit that transnational attachments are not binary opposites ([71]), and instead, people may feel varying degrees of attachments to more than one element.

The first element refers to the transitions that individuals go through. These transitions may be physical or part of the transnational imaginary. A person who has not engaged in a physical transition may have preconceived ideas of a given lifestyle “on the other side”. As such, the person may incline more towards an imaginary transition over a physical one. On the contrary, a person who engages in physical transition will most likely shed their imaginaries of the target location and its people quite quickly, inclining more towards the physical and away from the imaginary transition. Likewise, these transitions will depend largely on how often they occur, in which the person may incline more towards continuous or sporadic transitions.

The second element is closely connected to the first and refers to the networks that a person establishes. These linkages may be familial, social, or both. In this sense, the person may incline to one side of the continuum over the other, or place themselves more towards the center. According to the type of transitions (i.e., physical or imaginary), how often these transitions occur, and the networks that develop from such transitions, the person is also likely to develop a particular attachment towards a given culture and language.

The third element in the diagram denotes the attachments that a person may develop towards their sending or host culture, as well as their home or target language. In this sense, a person may feel greater attachment towards the culture “left back home” or the current and receiving culture. Similarly, this attachment may also relate to the language practices from home, or those related to the new setting. It is likely that if a person engages in continuous transitions, the attachments may be rather equal to both cultures and languages, resulting in a greater sense of belonging to both cultures and languages. On the other hand, if a person engages in sporadic transitions, the attachments are likely to incline to one culture and language over the other.

It is important to consider that while family migration backgrounds and histories and broader societal migration practices can play a crucial role in predisposing a person to also engage in similar migration practices, they do not define how a person will eventually engage in transnational migration.

Below this initial section of the diagram, *conocimiento* and identity are interrelated. I argue that transnationalism, *conocimiento*, and identity each influence one another, and thus, call for concurrence. In other words, the development of *conocimiento*, as well as the [re]construction of the identity of an individual, will be strongly related to the transitions, networks, and attachments of each individual. As each individual engages in a transition, develops and/or connects to a network, and [re]considers certain attachment to a given culture and language (or both), *conocimiento* and identity will be impacted and reconstructed. Consequently, this will develop and [re]construct what I refer to as transnational capital, which signifies a wealth of knowledge that allows one to shift between identities as needed in order to navigate social and professional involvement with the world and its people.

Despite the many similarities in the stories of those who engage in transnational migration, it is nearly impossible for two people to experience and be impacted by such a phenomenon in the same way. Thus, I align with [12] ([12]) and [87] ([87]), and also argue that transnationalism is lived differently by each individual. In turn, this phenomenon has a unique impact on each individual as well. That is, in spite of the many similarities in the experiences and negotiation of transnationalism among individuals who develop strong kinship, each will ultimately live their transnationalism process(es) in a rather unique way, and will most likely experience the unpredictability of settling in a given location with the wealth of knowledge accumulated through personal lived experiences.

Some people may engage in transnationalism from imaginary transitions by listening to the stories of their ancestors and simply envisioning people and lifestyles “on the other side”, even without physical transition. However, engaging in physical transition will be crucial in comparing and contrasting the illusions to the realities of life “*del otro lado*”. In this sense, individuals who engage in transnational migration are likely to almost always live in simultaneity, or live lifestyles that incorporate elements from both settings in a transnational manner ([73]). It is important to mention, however, that legal status is key in doing so, as it is a central determinant of how often a person may physically transition from one setting to another.

## 4. Conclusions

This ethnographic self-reconstruction centers my personal story, engaging in transnational migration practices. I began by presenting a broad introduction to the field of transnationalism. I followed by situating transnationalism within education, centering particularly on transnational children and youth and their identity and knowledge [re]construction processes. I then discussed the under-explored comparative stance on transnationalism and education. Following, I recalled my story of engagement in continuous transnational migration practices, [re]engaging in transitions, [re]creating networks, and [re]developing attachments, as well as reflecting on how these have and continue to mediate my identity [re]construction and *conocimiento*.

Through reflecting on my personal experience, I sought to highlight the challenges in addition to the many assets that many times are unseen and underacknowledged within diverse social and academic settings on both sides of the border. Strong kinship developed into a strong sentiment of cooperation to help navigate and behave accordingly within different social environments. In turn, this kinship motivated one to attempt to rediscover and reinvent oneself accordingly through navigating the “in-between”. Despite the portrayal of a rather successful transnational education experience, my lived experiences are quite unique, and also encompass an array of challenges with regard to uncertainties in navigational skills on both sides of the border. Nonetheless, these experiences led to my development of *conocimiento*, identity, and transnational capital. This is portrayed in my proposed framework, which seeks to counter “either/or” discourses and instead argues for a “both from here and from there” sense of belonging and attachment. As such, those who have engaged or are engaging in transnational migration practices may place themselves in different (and perhaps temporary) points along each of the continuums, but can rely on their skillsets and knowledge to shift to either end intentionally and/or strategically as needed.

Ultimately, I present my story to contest binaries in academic discourse, which may also continue to promote deficit views towards transnationals, emphasizing their lack of certain cultural and linguistic aspects, lack of a sense of belonging to both cultures and languages, and lack of overall understanding of maneuvering through society. I presented continuums with intentionality to argue that those who have engaged in transnational migration practices (even through the imaginary) have varying degrees of attachments to the several elements described in each of the continuums, never completely detaching themselves from either end of the various continuums. Hence, transnationalism is unique to each individual and shapes their worldviews, which can be [re]constructed accordingly. Moreover, one’s transnational capital, or the wealth of knowledge gained through engaging in transnational migration practices, will allow one to shift between and reconstruct identities and draw on one’s skillsets and ways of knowing to properly navigate within and among various social and professional domains.

## Figures and Tables

**Figure 1 behavsci-15-01539-f001:**
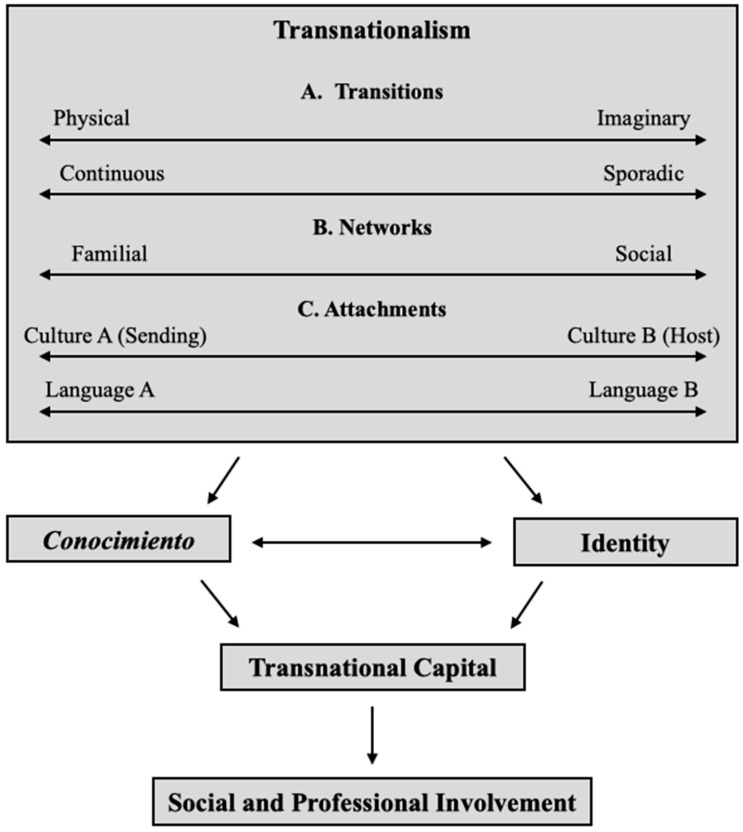
Framework for understanding transnationalism and its interrelated elements.

## Data Availability

The original contributions presented in this study are included in the article. Further inquiries can be directed to the corresponding author.

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
