# Peer review of "At the Intersection of Transnationalism, Identity, and Conocimiento: An Autoethnography"

_behavsci, 2025, doi:10.3390/bs15111539_

Round 1
Reviewer 1 Report
Comments and Suggestions for Authors
There is a lot here that is good and important and, in the big picture, I recommend this for publication. However, there are a number of vague or slightly misleading phrasings that merit attention before then. The author deserves praise for an amazingly thorough review of the literature, but that same thoroughness means we don't actually get to any of the autoethnographic parts until we have negotiated a dozen pages of dense reading. The current paper's structure does echo academic convention (with the lit review preceding the new research), but reader enthusiasm to persevere to the autoethnographic portion and the way they consume the lit review could both be enhanced if there were 2 or 3 sentences or even a short paragraph in the introduction that previewed some of the themes that the autoethnography was going to explore.
The opening sentence (p. 1, line 19) uses a citation from 2000 to say that movement of people, capital, products, etc is "incessant and ongoing." It's hard to pair a 25 year-old citation with the present tense "is". It's not a bad citation, but I would change the verbs and verb tenses used, perhaps something in the vein of "We have long known..." or "For more than a generation..." Those phrasings or something like that could still set up the author's point that a first-hand autoethnography is a valuable contribution to a dynamic that has long been studied other ways.
In the very next sentence (and reeling from the neo-colonialist as well as corrupt and nihilist policy frameworks of Trump 2.0) I am uncertain how apt it is to call the current era "post-colonial" (p. 1, line 20)
The present tense emphasis of "may better inform the current transnational reality" reminds me of Concha Delgado-Gaitan's (1990) caution against using the present tense with ethnography because it misconstrues the contextual and ephemeral as 'timeless'. Time-stamping this point instead by referencing specific events (like the US's return to Trump and/or Mexico's inauguration of its first woman president) would get away from the vagueness of "current." I am also convinced by the century-plus argument of anthropology that humanity is marked by ontological variation. So I would pluralize "reality" to "realities."
I would change "early on research on transnationalism" (p. 2 line 58), to "early on theorizing about transnationalism." One can readily place research on transnationalism well-before Kearney (1991), for example practically all early 20th Century research in America on immigration, but it is safer to locate the turn to theorizing transnationalism as more recent (with Kearney as a more apt citation in that framing). (The author himself seems to concede point on p. 3 lines 111-113)
While the LEP perspective is appropriately critiqued (p. 5, line 164) perhaps it should be briefly noted that that terminology came from the Lau v. Nichols (1974) Supreme Court decision that found that school systems that failed to attend to some students lack of English proficiency was problematic and that districts needed to identify who merited additional support. Absent an identification of a problem, there would be no mandate for attention to its negotiation/resolution.
If needed, the citation below names this problem: "An additional challenge arises as this population may be moving in and out of classifications that view them through a deficit perspective." (p. 5, line 175):
Hamann, E. T., & Reeves, J. (2013). Interrupting the Professional Schism that Allows Less Successful Educational Practices with ELLs to Persist. Theory Into Practice, 52(2): 81-88. http://dx.doi.org/10.1080/00405841.2013.770325 or https://digitalcommons.unl.edu/teachlearnfacpub/352/
As a key caveat, while all the students referenced by Zúñiga and Hamann (2015) have Mexican heritage (and Mexico-born parents), not all of the students were born in Mexico. So this line should be amended slightly: "The results depict the many challenges of inhabiting a territorial dislocation by children who migrate to Mexico, the country of their birth, for the first time (Zúñiga & Hamann, 2015)." (p. 7, lines 294-296)...Later (line 300), I wouldn't say that these children "are confronted with...a foreign culture." It's not entirely foreign. Perhaps 'an only semi-familiar culture' would be more apt , indicating that the familiarity is limited, but not non-existent. Also I would have this sentence by the final one of the previous paragraph and have the new paragraph start with the sentence (line 300) which begins: "Sierra and López (2013)..."
Particularly given that Baja California hosts the largest number of transnational students in Mexico and that practically all those transnational students are in the urban settings of Tijuana, Mexicali, and Ensenada, I don't think it's accurate anymore to say that transnational students "are likely to attend rural schools" (line 304). More accurately, they are likelier to attend rural schools than is the mononational Mexican student population. Like the population writ large, the latter is now decidedly more numerous in urban schools, while the transnational student population is too, but not quite as much.
I would edit this line "Most educators do not speak any English, or have relatively low competence in the language (Mexicanos Primero, 2015)" (line 331) to instead just say: "Most educators have relatively low competence with the English language (Mexicanos Primero, 2015)" and perhaps would add "and don't use it instructionally."
This point is true: "Also, educators are largely unfamiliar with the U.S. education system and curricula, which contributes further to schooling complexities and higher instances of failure." (lines 332 and 333), but it is worth noting several efforts to change this, including substantial investment in the PROBEM program and previously in the Educación sin Fronteras initiative (2006-2012). As examples of Mexican supported efforts to build Mexican teachers readiness for transnational students see:
Hamann, E. T., Zúñiga, V., & Sánchez García, J. (Eds.) (2022). Lo que conviene que los maestros mexicanos conozcan sobre la educación básica en Estados Unidos [What Mexican Teachers Need to Know About ‘Educación Básica’ in the United States]. Universidad Autónoma de Nuevo León Press. https://editorialuniversitariauanl.publica.la/library/publication/lo-que-los-maestros-mexicanos-conviene-que-conozcan-sobre-la-educacion-en-estados-unidos
Sánchez García, J., Zúñiga, V., & Hamann, E. T. (2010). Guía didáctica: Alumnos transnacionales. Las escuelas mexicanas frente a la globalización. Secretaria de Educación Pública [Mexico].
Zúñiga, V., Hamann, E. T., & Sánchez García, J. (2008). Alumnos transfronterizos: Las escuelas mexicanas frente a la globalización. Secretaria de Educación Pública [Mexico].
I would change "prevent" (line 337) to "inhibit." There are examples of successful transnational students in Mexico.
Hamann is misspelled in line 381.
The transnational students in higher education noted by Cortez Roman and Hamann (2014) had mostly not been deported (despite line 432), but they had been largely blocked from accessing higher education in Arizona because of a change in that state's law.
Relevant to section 2.4 (pp. 11-13), Juan Guerra (1999) compellingly used the term "transnational community" (emphasis on the singular) in his examination of who had highest status/greatest social capital in a studied speech-community of 68 people with life experience in Chicago and two Mexican communities. That's the earliest assertion of 'transnational knowledge' that I have encountered that uses the term 'transnational' (although there likely are some even earlier than that). His work also would position the author to move away from the old migration studies trope of 'sending' and 'receiving' communities to instead use the more neutral 'previous' and 'current' to reference geolocation.
The points made about Petrón and Greybeck's (2014) findings (line 541 to 553) seem to align with Guarnizo and Smith's (1998) theoretical claims regarding a 'transnationalism from below'. Given that Guarnizo and Smith's work is already cited in other parts of the manuscript, perhaps a quick tieoin here too would be expedient.
In line 563 the word "adopting" is used when I think the intended term was "adapting."
I would add the caveat 'can' to this sentence (lines 620-621): "Hamann and Zúñiga (2011b) depict the estrangements that transnational children CAN face on both sides of the U.S.-Mexico border as a result of continuous migration practices."
The "everyday ruptures" referred to in Hamann and Zúñiga (2011b) do not really precipitate migration (as implied in line 623), but rather are part of the bumpiness that transnational students need to routinely negotiate because of their changed national educational contexts.
In line 696 I think "singled out" is intended (not "signaled out")
I think "seventh grade" gets erroneously repeated in the parenthetical asides in lines 709-710. I think the full-year is supposed to be "eighth grade", unless the author is trying to tell us that he had to repeat a year (which I don't think is the case)
This turn of phrase is important but vague: "but was faced with the process of apostilles" (line 733). I think the author means that his registration and enrollment in preparatoria in Mexico was delayed because his transcripts and other placement documents from the US needed to be made official by using the apostille process (like the notary process) to verify the legitimacy of the documentation from US schools.
Because US high school (grades 9-12) and Mexican preparatoria (grades 10-12) aren't exact matches for each other, the use of 'high school' to refer to preparatoria in line 735 gets a bit confusing. The reason 10th grade was the first or lowest grade level in the school was because it was a Guanajuato prepa not a US high school that was being enrolled in.
I wonder if line 764 (p. 16) would be a bit clearer if "this group of students" was changed to "our group of English pre-service teachers." If that change makes sense, then all the 'their's that follow in the remainder of the paragraph should also be switched to the first person plural 'our'.
Given that this is supposed to be autoethnographic (and the author describes his transnational experience), the continued use of 'they/their', instead of us/our in the middle paragraph on p. 16 (lines 768-774) gets confusing.
I know that ELT (e.g., line 778) refers to English language teaching, but I don't think it is ever defined as such.
Would it fit to add the words 'youth' or 'student' to the following sentence: "Through my dissertation study, I sought to better understand YOUTH/STUDENT transnationalism, what it entails and what the practices are, as well as highlight the many assets that transnational STUDENTS WHO BECOME teachers may bring into the classroom as a result of their experiences" (lines 788-790)?
Add 'other' to this sentence (line 795): "In reflecting on my own life story and those of OTHER individuals who have engaged in migratory movements among and between two different geographical settings and cultural contexts, I find it useful to also draw on interrelated ideas emerging from the literature on transitions occurred through the life course of a given person.
While I like Figure 1 (p. 18) broadly, I would remove the 'sending' and 'host' terminology, instead replacing it with the more descriptively neutral 'previous' and 'new'. It also seems worth highlighting that 'conocimiento' is left in Spanish rather than translated (presumably because there isn't a particularly precise/apt translation) which highlights the value of knowing and being able to use both languages.
This point (lines 867-869) is no doubt true: "Thus, I align with Casinader (2017) and Mora Vázquez et al. (2018), and also argue that transnationalism is lived differently by each individual. In turn, this phenomenon has a unique impact on each individual as well." Yet it seems to ignore another key point of the paper that there are similarities in students' negotiation of transnationalism. Hence the affinity towards each other as undergraduates in the ELT program. The similarities are as salient as the points of uniqueness.
The concluding section is too brief and, crucially, it never asserts why an autoethnographic perspective added value to the paper. (A strong argument can be made for how it added value, but that is not directly considered.) Absent autoethnography, how would the author have known of the prospective value of parent emphasis on strong academic performance? Or the value of a 'Spanish while at home' policy in Oregon which meant comprehending Spanish in Mexican schools was not as hard as it is for some transnational students?
As a broader and more random note, Arjun Appadurai has been an important and long-time theorizer of transnationalism (albeit not between the US and Mexico per se) and his absence from the text and list of citations is a bit puzzling, given how otherwise thorough the author's reading appears to be.
Also, it is worth somewhere acknowledging that the author's own transnational education experience and those of other transnational students have been relatively successful. The author does have insight from his biography and from the biography of peers he studied with in university about various challenges that transnational students negotiate, but those challenges were not enough to lead to school failure and abandonment. As such, this account better captures the experience of those who succeeded than those who were overwhelmed by their attempted negotiation of the two systems.
Finally, my own work on transnationalism has been critiqued really as a study of binationalism. This author may want to briefly consider why transnationalism is a better term for his experience than binationalism.
Author Response
|
Reviewer 1 |
|
|
Recommended Revision |
Response and Revision |
|
Introduce 2 or 3 sentences or a short paragraph in the introduction to preview some of the themes that the autoethnography will explore. |
I added a paragraph at the end of the introduction section as a sort of roadmap to preview the content of the paper. |
|
Reword opening sentence (p. 1, line 19) to set up author’s point that topic has been studied other ways. |
The sentence has been changed as per the suggestion. |
|
Reconsider “post-colonial” (p. 1, line 20). |
The word was omitted. |
|
Caution of using present tense, get away from vagueness of using “current”. |
The wording and verb tenses have been reconsidered. |
|
Change “early on research on transnationalism” (p.2, line 58) to “early on theorizing about transnationalism”. |
The suggestion has been considered and the wording has been modified. |
|
Amend line about Zúñiga & Hamann (2015) regarding Mexican heritage students (p. 7, lines 294-296). |
The suggestion has been considered, and the sentence has been modified and moved to the end of the previous paragraph. |
|
Reconsider the accuracy of saying transnational students “are likely to attend rural schools” (line 304). |
The wording has been modified accordingly. |
|
Rephrase “most educators do not speak any English, or have relatively low competence in the language (Mexicanos Primero, 2015) (line 331). |
The sentence has been edited accordingly. |
|
Reword “educators are largely unfamiliar with the U.S. education system and curricula, which contributes further to schooling complexities and higher instances of failure” lines 332 and 333) with suggested literature. |
The suggested literature has been added. |
|
Change “prevent” to “inhibit” (line 337). |
The wording has been changed. |
|
Hamann is misspelled in line 381. |
The spelling has been corrected. |
|
Change “adopting” to “adapting” (line 563). |
Wording has been changed. |
|
Add “can” to Hamann and Zúñiga’s (2011b) work (lines 620-621). |
Wording added to the manuscript. |
|
Change “signaled out” to “singled out” (line 696). |
Wording changed accordingly. |
|
Reconsider “seventh” and “eighth” grade (lines 709-710. |
Correction made. |
|
Rephrase wording around apostilles (line 733). |
Wording has been changed accordingly. |
|
Reword “high school” and “preparatoria” (line 735) as they aren’t exact matches. |
The corresponding changes have been made. |
|
Reconsider discourse around “this group of students” to “our group of English pre-service teachers” (line 764, p. 16). |
Rewording to the section has been made accordingly. |
|
Define ELT earlier on in the manuscript. |
ELT has been defined. |
|
Add “other” to line 795. |
Wording has been added. |
|
Consider changing “sending” and “host” terminology to “previous” and “new” on figure 1 (p.18). |
I would like to keep wording as is. I argue that the “host” setting may not always be “new” and/or unfamiliar. It may be rather temporal, but nonetheless functions as the “host” setting for a given duration of time. |
|
Reconsider arguments in lines 867-869 around transnationalism being lived differently by each individual. |
The argument has been further developed. |
|
The concluding section is too brief. Assert why an autoethnographic perspective added value to the paper. |
The conclusion has been expanded as suggested. |
|
It is worth acknowledging that the author’s own education experience and those of others have been relatively successful. |
I acknowledge my education experiences and highlight the importance of kinship in navigating with relative success. |
Reviewer 2 Report
Comments and Suggestions for Authors
This article makes a significant contribution to the field, particularly given contemporary U.S. migration dynamics characterised by deportations to Latin American countries, and also the continue phenomenon of multigenerational transit migration.
The article is well-written and addresses a topic of considerable scholarly importance that merits publication. The author demonstrates strong engagement with existing literature, providing extensive and rich references to both theoretical frameworks and empirical studies throughout the literature review and subsequent analysis. These references effectively contextualise the study and add credibility to the author's conclusions.
A key strength of this work is the author's references on the importance of reforming education systems on both sides of the border. The author compellingly argues that students—regardless of age or grade level—require comprehensive support during their adaptation processes and throughout their educational, social, and cultural transitions in either country.
The author identifies a critical gap in the existing scholarship: the scarcity of studies examining transnational educational practices across borders, particularly those employing autoethnography as a methodological approach.
I concur with the author's assertion that these transnational experiences are inherently unique and context-dependent. The autoethnography methodology demonstrated by the author is particularly well-suited to capturing the nuanced, lived dimensions of cross-border educational experiences that quantitative or purely theoretical approaches might overlook.
This article offers a refreshing perspective that prompted reflection on the broader conditions of migrants in transit globally.
Please revise the reference list to conform to APA citation style, which is required by this journal. The reference list is correctly cited, but it does not need to be numbered, and each reference should be indented. The author may find it helpful to review the reference formatting in other recent publications from this journal as a guide.
Author Response
|
Reviewer 2 |
|
|
Consider suggestions from Reviewer 1. |
All suggestions have been considered. |
|
Revise reference list to conform to APA. |
APA list has been revised accordingly. |